# Damping Characteristics of a Novel Bellows Viscous Damper

**DOI:** 10.3390/s24196265

**Published:** 2024-09-27

**Authors:** Yang Chen, Chao Qin, Honghai Zhou, Zhenbang Xu, Anpeng Xu, Hang Li

**Affiliations:** 1Changchun Institute of Optics, Fine Mechanics and Physics, Chinese Academy of Sciences, Changchun 130033, China; chenyang21@mails.ucas.ac.cn (Y.C.); zhouhonghai21@mails.ucas.ac.cn (H.Z.); xuzhenbang@ciomp.ac.cn (Z.X.); xuanpeng20@mails.ucas.ac.cn (A.X.); lixing20@mails.jlu.edu.cn (H.L.); 2University of Chinese Academy of Sciences, Beijing 100049, China

**Keywords:** bellows-type fluid viscous damper, vibration isolation, nonlinear damping, micro-vibration

## Abstract

Micro-vibrations during the operation of space remote sensing equipment can significantly affect optical imaging quality. To address this issue, a bellows-type viscous damper serves as an effective passive damping and vibration isolation solution. This paper introduces a bellows-type viscous damper with adjustable damping capabilities, designed for mid- to high-frequency applications. We developed a system damping model based on hydraulic fluid dynamics to examine how different factors—such as viscous coefficients, damping hole lengths, hole diameters, chamber pressures, and volumes—influence the damping characteristics. To validate the theoretical model, we constructed an experimental platform. The experimental results show that the theoretical damping curves closely match the measured data. Moreover, increasing the chamber pressure effectively enhances the damper’s damping coefficient, with the deviation from theoretical predictions being approximately 4%.

## 1. Introduction

Recently, optical remote sensing payloads have played a pivotal role in space exploration and ground-based remote sensing detection, with imaging resolution and stability becoming key design metrics [1,2,3]. Micro-vibrations caused by the operation of internal fast mirrors, chillers, and environmental perturbations significantly affect the imaging quality of these payloads. These micro-vibrations typically occur within the frequency range of 0.1 Hz to 1 kHz, with amplitudes in the micrometer scale. To improve imaging quality, vibration isolation is essential [4,5]. Damped vibration isolation systems are generally classified into passive, active, and semi-active. Passive damping methods include friction damping, viscous damping, viscoelastic damping, and particle damping [6,7,8]. To enhance damping performance and broaden application scenarios, combined dampers have been developed. Su et al. designed a twisted-rope fluid-viscous damper that combines friction and viscous damping, achieving higher damping characteristics than traditional designs [9]. In addition, Fakhraei et al. conducted an in-depth study on the nonlinear dynamic behavior of suspension systems equipped with viscous and magnetorheological dampers under continuous road conditions, revealing a significant effect of chaotic behavior on ride comfort and load distribution. These findings provide new ideas for improving damping performance in complex systems [10]. Advances in new materials have led to the development of memory alloy (SMA) slip friction dampers and metal dampers with higher energy dissipation efficiency [11,12]. Active damping is generally realized either through the parallel connection of an actuator and a passive damper or by using driving power with active control algorithms. To reduce driving power and utilize vibration energy, semi-active dampers have been rapidly developed [13]. Bhowmik et al. demonstrated effective control using the linear quadratic Gaussian (LQG) control of magnetorheological damping with only a 2.5 V supply [14]. The selection of vibration isolation dampers depends on their specific characteristics and requirements of the application and environment.

In the field of vibration isolation technology, quasi-zero stiffness (QZS) techniques have increasingly emerged as crucial methods for addressing micro-vibration issues [15,16,17,18]. To enhance the imaging stability of optical remote sensing payloads in complex environments, researchers have extensively investigated the performance of QZS systems. Zhao et al. proposed a lightweight and high-performance beam-like unit cell 3D metamaterial QZS designed to overcome the limitations of traditional QZS systems within specific frequency ranges. The experimental results indicate that this system effectively attenuates up to 90% of vibration amplitudes within the 100–600 Hz frequency range, thereby significantly improving the isolation performance of the system [19]. To further enhance the vibration isolation effectiveness of QZS systems, Wen et al. developed a QZS isolation system incorporating shear-thickening viscous dampers (SVDs), achieving lower initial isolation frequencies and peaks, and demonstrating excellent isolation capabilities in the mid- to high-frequency range [20]. Meanwhile, nonlinear energy sink (NES) technology has gained increasing attention from researchers due to its unique energy directional transfer properties [21,22,23]. Zhao et al. applied inertial NES to vibration control in marine structures, finding that it offered significant advantages over traditionally tuned mass dampers (TMDs), particularly in complex vibration environments, where it demonstrated enhanced damping performance and stability [24].

Fluid viscous dampers (FVDs), the most prevalent means of passive vibration suppression, are widely used in bridge construction, marine, and aerospace applications [25,26,27,28,29]. Applications such as those in the Giant Magellan Telescope (GMT) demonstrate their important role in high-precision equipment [30]. To identify the parameters and model the FVD system, Greco et al. [31] and Vasile et al. [32] proposed a new method based on the Kelvin-Voigt rheological model, achieving better results. For extreme working conditions, Chmielowiec et al. [33] and Hu et al. [34] investigated the dynamic behavior of FVDs under leakage and lack of liquid. The thermal effects of a prolonged damper operation are also a concern. Lak et al. established an energy–temperature curve to estimate thermal behavior and avoid the risk of system overheating [35]. Additionally, rotating fluid dampers (RFDs) are crucial in applications such as the vibration damping of offshore platforms due to their low cost and high damping efficiency [36].

Compared to other passive dampers, bellows-type viscous dampers, characterized by their high energy dissipation density and minimal influence from the external environment, have been successfully validated for micro-vibration isolation in space loads [37,38]. Scholars have extensively researched bellows-type FVDs. Jiao et al. proposed a simplified model for bellows-type dampers under medium–high excitation to analyze changes in nonlinear stiffness and damping; however, the accuracy of its nonlinear fitting at low viscosity requires further verification [39]. To enhance low-frequency vibration isolation performance, Teng et al. incorporated additional inertial mass into a typical bellows-type FVD structure, achieving a leftward shift in the system’s vibration isolation curve and demonstrating high static and low dynamic stiffness characteristics [40]. To improve the understanding of bellows-type FVDs, Oh et al. proposed a three-parameter model, treating the stiffness generated by internal fluid compression as a constant, which reduced its accuracy [41]. To improve the model’s accuracy, Jiao et al. used an approximate analytical modeling method to establish a reduced-order model, accounting for internal hydraulic stiffness changes and bellows volume deformation, resulting in a computational error of less than 8.2% from 1 to 300 Hz [42]. Despite abundant research on bellows-type dampers, it tends to focus on the lower-frequency band, lacking discussion and reports on high-frequency vibrations above 300 Hz.

To meet this requirement, this paper proposes a design for a new type of adjustable-damped bellows-type viscous damper, investigates its nonlinear damping characteristics under high-frequency harmonic perturbation, and studies the effects of different design parameters on the system’s damping performance. The details are as follows: the system components and structural characteristics of the damper are described in Section 2, and the damping model is derived in Section 3, where the effects of different design parameters on the damping model are analyzed. Finally, a comprehensive experimental platform is constructed in Section 4 to analyze and verify the validity of the theoretical model using the experimental results obtained.

## 2. System Structure

The structure of the adjustable damping bellows-type liquid damper designed in this paper is shown in Figure 1, in which the bellows and flange are connected in series with each other to form the cavity and flow path of the damper, and the damping liquid flows between the cavities and flows through the damping holes to dissipate energy. The overall structure is built into the flange cover; one end is fixed on the flange cover by screws, and the other end is pressed by the internal pressure adjustment structure which is composed of screws and the flange. The screw in the rotating internal pressure adjustment structure drives the damping cavity to perform tensile and compressive movements to change its internal pressure. At the same time, in order to ensure that the moving parts are strictly aligned along the axial translation, the flange and the cavity connection are equipped with a precision guiding diaphragm and are fixed by the flange cover. The claw structure of the flange passes through a gap in the flange cover and is connected to the end cap. To ensure that the chamber is airtight and to reduce the air content of the damping fluid, a small vacuum oiling truck is used to oil the damper through quick change coupling. The external vibration is transmitted to the flange through the end cap, and the movement of the flange drives the internal damping fluid to flow along the gap between the upper and lower chambers, and the energy is dissipated by the friction between the fluids.

Given that the damper designed in this study is intended to function effectively across a broad frequency spectrum, a thorough modal analysis of its structural behavior is crucial. The first six natural modes of the damper, depicted in Figure 2, provide key insights into its dynamic characteristics. To ensure optimal performance and to prevent any potential resonance-related issues, it is imperative to avoid operational frequencies near these modes and their corresponding regions. Failure to do so could compromise the dampers’ effectiveness, leading to undesirable vibrations or mechanical instability.

## 3. Modeling and Characterization of Bellows-Type Viscous Dampers

### 3.1. Model Based on Hydraulic Fluid Dynamics

When the frequency of environmental excitation is in the lower-frequency band, fluid compression is relatively small, allowing the fluid to flow smoothly through the damping holes. In this scenario, the hydrodynamic damping model can accurately describe the dampers’ performance characteristics. However, as the excitation frequency gradually increases, the amount of compressed fluid in the cavity rises significantly, leading to a reduction in flow through the damping holes. Consequently, the damping characteristics of the damper change markedly with increasing frequency, necessitating a more precise hydrodynamic model to accurately describe its behavior.

In this paper, the damping fluid was modeled as a compressible elastomer, and the damping model for the bellows-type fluid damper was developed using hydraulic fluid dynamics theory. Figure 3 illustrates the schematic diagram of the damping element of the bellows-type fluid damper. The flange inside the damper divides the space into two uniform chambers. When the flange is subjected to sinusoidal excitation, it undergoes a reciprocating motion, causing the damping fluid to flow back and forth between the two chambers through the damping orifices. During this process, part of the fluid flows directly through the damping orifices to the other chamber, while the remaining fluid is compressed under the chamber’s pressure. The motion of the fluid between the two chambers is described by the following differential equation.
(1)$$\left\{\begin{array}{l} V_{1}:A_{P}v=Q+V_{1}P_{1}^{\prime }/\beta_{1}\\ V_{2}:A_{P}v=Q-V_{2}P_{2}^{\prime }/\beta_{2} \end{array}\right.$$

*V*_1_ and *V*_2_ represent the volumes of the left and right chambers, respectively. *β*_1_ and *β*_2_ denote the bulk moduli of the liquid at pressures *P*_1_ and *P*_2_, respectively. *Q* indicates the flow rate through the damping hole. *A_p_* is the cross-sectional area of the cavity, *A_g_* is the cross-sectional area of the damping holes, and *v* is the velocity of the flange movement.

The vibration excitation amplitude of medium and high frequency is generally micron scale, relative to the size of the damper; this amplitude caused by the change in the volume of the damper chamber is negligible. Therefore, when the flange is in reciprocating motion, it can be assumed that the volume of the two chambers does not change, i.e.,
(2)$$V_{1}=V_{2},\beta_{1}=\beta_{2}$$

The vicious loss of fluid flowing in a damped orifice can be described by liquid resistance.
(3)$$R_{H}=\frac{\Delta P}{Q}=\frac{8\mu l}{\pi r_{0}^{4}}$$

*R_H_* is the hydraulic resistance. Its magnitude is related to the dynamic viscosity *μ* of the damping fluid, the length *l* of the damping hole, and the radius *r*_0_ of the damping hole. The fluid is laminar when it flows through the damping hole, and the flange drives the two chambers to produce a differential pressure force equal to the sum of the inertial force of the fluid flow and the damping force produced during the flow, which are both represented by the liquid pressure equilibrium equation:(4)$$(P_{1}-P_{2})A_{g}=\rho L_{g}\dot{Q}+R_{h}QA_{g}$$

*ρ* is the density of the damping fluid and *L_g_* is the length of the cavity. Combining Equation (3) with Equation (4) forms the following coupled equation:(5)$$\left\{\begin{array}{l} {\dot{P}}_{1}=\frac{-Q\beta }{V_{0}}+\frac{A_{P}v\beta }{V_{0}}\\ {\dot{P}}_{2}=\frac{Q\beta }{V_{0}}-\frac{A_{P}v\beta }{V_{0}}\\ \dot{Q}=\frac{P_{1}A_{g}}{\rho L_{g}}-\frac{P_{2}A_{g}}{\rho L_{g}}-\frac{R_{h}QA_{g}}{\rho L_{g}} \end{array}\right.$$

Based on the previous force analysis of the piston, the dynamical equations can be written in the form of a state space. The equation of the state of the system can be written as
(6)$$\left\{\begin{array}{l} \dot{x}=Ax+Bv\\ F=cx \end{array}\right.$$
where
(7)$$\begin{array}{c} x=\left[\begin{array}{c} P_{1}\\ P_{2}\\ Q \end{array}\right],A=\left[\begin{array}{ccc} 0 & 0 & -\beta /V_{1}\\ 0 & 0 & \beta /V_{2}\\ A_{g}/\rho L_{g} & -A_{g}/\rho L_{g} & -R_{H}A_{g}/\rho L_{g} \end{array}\right]\\ B=\left[\begin{array}{c} \beta A_{p}/V_{1}\\ -\beta A_{p}/V_{2}\\ 0 \end{array}\right]\\ C=A_{P}[1,-1,0]\\ F=(P_{1}-P_{2})A_{P} \end{array}$$

Assuming that the initial conditions of the corresponding variables are 0, and considering the general case *d* = 0, the Raschner transform of the above equation is given by



(8)
$$\begin{array}{l} sx(s)=Ax(s)+Bv(s)\\ F(s)=cx(s) \end{array}$$



The transfer function of the damping force for the excitation velocity is obtained:(9)$$G\left(S\right)=\frac{F(S)}{V(S)}=C{\left[SI-A\right]}^{-1}B$$

As the external excitation is a sinusoidal displacement excitation, the displacement and velocity can be expressed as
(10)$$\begin{array}{l} x=x_{0}\sin(\omega t)\\ v=\omega x_{0}\cos(\omega t) \end{array}$$

*x*_0_ is the excitation amplitude. The velocity after the Laplace transform is expressed as
(11)$$v(s)=\frac{x_{0}\omega s}{s^{2}+\omega^{2}}$$

As the external excitation is a sinusoidal displacement excitation, the output force in phase with the displacement is elastic, and in phase with the velocity it is the damping force, giving the damping force as
(12)$$F_{c}=\frac{A_{2}A_{3}k_{0}\omega x_{0}\cos(\omega t)}{A_{2}^{2}\omega^{2}+A_{3}^{2}-2A_{3}\omega^{2}+\omega^{4}}$$
(13)$$A_{1}=A_{2}=\frac{R_{h}A_{g}}{\rho L_{g}},A_{3}=\frac{2A_{g}\beta }{\rho L_{g}V_{0}},k_{0}=\frac{2\beta A_{p}}{V_{0}}$$
where *F_c_* is the damping force. The value of the damping coefficient *c* is
(14)$$c=\frac{F_{c}}{v}=\frac{A_{2}\cdot A_{3}\cdot k_{0}}{A_{2}^{2}\omega^{2}+A_{3}^{2}-2A_{3}\omega^{2}+\omega^{4}}$$

It is observed that the damping coefficient *c* is related to the cross-sectional area of the damping holes *A_g_*, the length of the holes *L_g_*, the diameter of the holes *r*_0_, the viscosity coefficient μ, the density of the damping fluid *ρ*, the bulk modulus of the fluid *β*, and the volume of the cavity *V*_0_. The variation in these parameters affects the damping effect of the damper.

### 3.2. Effect of Different Parameters on Damping Characteristics

In bellows-type viscous fluid dampers, the damping coefficient is significantly influenced by several design parameters. Thus, it is crucial to investigate how different design parameters affect the damper system. Figure 4a illustrates the variation in system damping with frequency for various damping fluid viscosities. The figure shows that there is a peak value in the damping coefficient. For instance, with a damping fluid viscosity of 3000 cst, the damping coefficient initially increases with frequency. After reaching a peak value, it gradually decreases as the frequency continues to rise. This behavior is due to the energy loss associated with fluid flow through the damping orifice. The magnitude of this energy loss depends on the flow rate through the orifice and the fluid resistance. At low excitation frequencies, most of the fluid within the stroke of the damper flange can flow through the damping orifice. As the frequency increases, both the flow through the orifice and the damping coefficient increase. However, as the frequency continues to rise, the cavity pressure increases, compressing the fluid further. Eventually, this causes the flow through the orifice to decrease and the damping coefficient to diminish. Overall, the damping coefficient exhibits an increase followed by a decrease with frequency, resulting in a peak damping value. Higher viscosities in the damping fluid lead to a more pronounced damping effect in the low-frequency range and a more stable overall damping value. Conversely, lower viscosities enhance the damping effect near the peak frequency.

Figure 4b illustrates the variation in damping with frequency for different aperture diameters. The figure reveals that at lower frequencies, a smaller aperture diameter results in a larger damping coefficient. This is because increasing the aperture diameter significantly reduces fluid resistance, leading to a decrease in the damping coefficient at lower frequencies. Conversely, at higher frequencies, a larger aperture diameter allows more fluid to flow through the damping hole, thereby increasing the damping coefficient. Thus, smaller apertures are more effective for damping at low frequencies, while larger apertures enhance the damping effect near the peak frequency.

Figure 5a shows how damping varies with frequency for different damping hole lengths. At lower frequencies, shorter damping hole lengths result in higher damping coefficients. This is because, at lower frequencies, the flow rate of the damping fluid through the hole is similar across different hole lengths. However, increasing the hole length significantly increases fluid resistance, leading to a higher damping coefficient. At higher frequencies, longer hole lengths result in greater fluid resistance, causing more fluid to be compressed within the stroke of the damping piston and reducing the flow through the damping hole. Consequently, the damping coefficient decreases with increasing hole length. Therefore, shorter hole lengths are more effective for damping at low frequencies, while longer hole lengths enhance the damping effect at high frequencies.

Figure 5b illustrates how damping varies with frequency for different cavity volumes. Since changes in cavity volume do not affect fluid resistance, the primary factor influencing the damping coefficient is the flow rate through the damping orifice. Larger cavity volumes result in a smaller proportion of fluid within the damping orifice relative to the total volume, which makes the fluid more easily compressible. Conversely, smaller cavity volumes lead to a larger proportion of fluid in the damping orifice, increasing the damping peak value and enhancing the overall damping effect. Thus, smaller cavity volumes provide a broader range of high damping and a more effective overall damping performance.

Figure 6 shows how damping varies with frequency for different cavity pressures. Since changes in cavity pressure do not affect fluid resistance, the primary factor influencing the magnitude of the damping coefficient is the flow rate through the damping holes. As cavity pressure increases, the bulk modulus of the fluid also increases, making the fluid more resistant to compression. At low frequencies, the flow rate of the damping fluid through the holes remains similar across different cavity pressures, and fluid resistance is consistent; thus, cavity pressure has minimal impact on damping. However, at higher frequencies, the increased flow rate through the damping holes due to higher cavity pressures results in a larger damping coefficient. Therefore, higher cavity pressures lead to a larger damping peak and a more pronounced overall damping effect.

## 4. Experiments and Discussion

### 4.1. Experimental System Construction and Process

Figure 7 illustrates the structural composition of the entire test system, which included a shaker, damper, force sensor, acceleration sensor, displacement sensor, controller, data collector, and experimental tooling. The experimental tooling comprised the platform base, shaker base, anti-slope bearing, displacement sensor fixture, and lifting ring. The platform base was mounted on the test bench, and the shaker base, which can be adjusted in height, was secured to the platform base. The displacement sensor fixture attached the displacement sensor to the platform base, while the damper was fixed to the anti-slope bearing. A force sensor was connected in series between the shaker and the damper to measure the shaker’s output force. The acceleration sensor was mounted on the damper’s end cap to monitor acceleration changes. The damper was suspended from the anti-slope bearing by screws to facilitate gravity unloading. It was ensured that the entire experimental setup was aligned horizontally. The screw in the internal pressure adjustment mechanism was rotated to drive the damping cavity through tensile and compressive movements, adjusting the cavity pressure and modifying the liquid’s bulk modulus to alter the damper’s damping effect. The chamber pressure was calculated based on changes in chamber volume, with the internal pressure adjustment screw set to a predetermined position to complete the experimental setup.

Figure 8 illustrates the experimental procedure. A sinusoidal scanning signal from the experimental signal generator was amplified by a power amplifier and fed into the shaker, exciting the entire system. Simultaneously, signals from the force sensor, acceleration sensor, and displacement sensor were synchronized with data acquisition through the data collector. The measured acceleration data were then processed using frequency domain integration to convert them into displacement information. This allowed for the determination of the force–displacement relationship at each frequency and the plotting of the hysteresis loop curve. The damping value at each frequency was calculated using viscoelastic theory. The input frequency of the shaker was varied according to an incremental scheme, and the experiment was repeated to obtain the damping–frequency curve for each pressure setting. The damper’s compression-damping liquid cavity was then adjusted to change the cavity pressure, and the experiment was repeated to determine the damping coefficient as a function of cavity pressure.

### 4.2. Analysis of Experimental Results

The hysteresis loop curves were approximated and fitted to the measured data to determine the damping coefficients of the damper at the current excitation level. The input frequency of the shaker was then varied incrementally to obtain hysteresis loop curves at selected frequencies, as illustrated in Figure 9. The damping coefficients corresponding to each hysteresis loop curve were calculated, and the results were fitted to produce damping versus frequency curves. These curves, measured at various frequencies, are shown in Figure 10a.

Comparing Figure 10a with the resonant modal frequency of the damper reveals that the damping coefficient decreases sharply at the resonance point. This drop is due to the system’s enhanced vibration when the system’s frequency is close to the frequency of external excitation. In the resonant state, energy is transferred from the external excitation to the system more efficiently, propagating throughout the system. This enhanced energy transfer occurs because the system’s natural frequency is nearly identical to the excitation frequency, enabling the system to absorb and respond to the excitation more effectively. At resonance, the system can store more energy in its vibration modes, resulting in increased amplitude. During the experimental tests, the damper’s displacement results from both structural elastic deformation and system motion. Consequently, the measured data do not accurately reflect the damping coefficient at resonance.

To ensure the precision of the damping coefficient measurements, data points near the modal frequencies were excluded, and the resulting graphs were re-plotted. Near modal frequencies, significant resonance effects can occur, leading to a pronounced amplification of the vibration response. Such resonance phenomena introduce nonlinear distortions that compromise the accuracy of the damping coefficient measurements. Specifically, the concentration of vibrational energy at these frequencies results in extreme response amplitudes, which deviate from the true damping characteristics of the system. Consequently, excluding measurements within these resonance-affected frequency ranges is crucial to obtain reliable data. Figure 10b presents a comparison between the optimized experimental damping curves and the theoretical curves. The alignment of trends between the two validates the effectiveness of our approach to mitigate resonance effects, thereby providing more accurate experimental data and reinforcing the applicability of the theoretical model for damper design.

To assess the effect of cavity pressure on the damper’s damping coefficient, the damping fluid cavity was compressed using an internal pressure adjustment mechanism. The damping measurement experiment was then repeated to obtain damping curves under various pressures as a function of frequency, as shown in Figure 11. Comparing the damping coefficients at different cavity pressures reveals that increasing the pressure effectively enhances the damping coefficient of the damper. As indicated in Table 1, the damping coefficient increases with pressure, and the error between the experimental data and theoretical calculations remains within 4%.

## 5. Conclusions

In this paper, a novel bellows-type viscous fluid damper with adjustable damping is introduced and its damping characteristics over a wide frequency range are investigated. To address the inherent limitations of conventional fluid dynamic damping models, especially when describing the damping mechanism of viscous fluid dampers under broadband excitation, a simplified model based on hydraulic fluid dynamics is proposed, which considers the damping fluid as a compressible elastic body.

A well-designed experimental platform is constructed to empirically validate the proposed model. The experimental procedure involves a stepwise variation in the input frequency of the vibrator to generate damping curves at different pressures. Adjustments to the damper structure allow for the compression of the damping fluid cavity and modification of the cavity pressure, enabling repeated measurements of the damping coefficient as a function of cavity pressure.

It is shown that structural resonances within the damper significantly affect the accuracy of broadband damping measurements. Comparative analysis of the experimental data with theoretical predictions shows a high correlation, thus validating the theoretical model. The bellows-type viscous fluid damper exhibits pronounced nonlinear damping characteristics over a wide frequency range. Furthermore, increasing the cavity pressure significantly improves the performance of the damper. The deviation between the experimental data and the theoretical predictions is kept within 4%, highlighting the robustness and accuracy of the model.

Overall, this study contributes to the understanding of the damping behavior of bellows-type viscous fluid dampers and provides a validated theoretical framework for predicting their performance under different operating conditions.

## Figures and Tables

**Figure 1 sensors-24-06265-f001:**
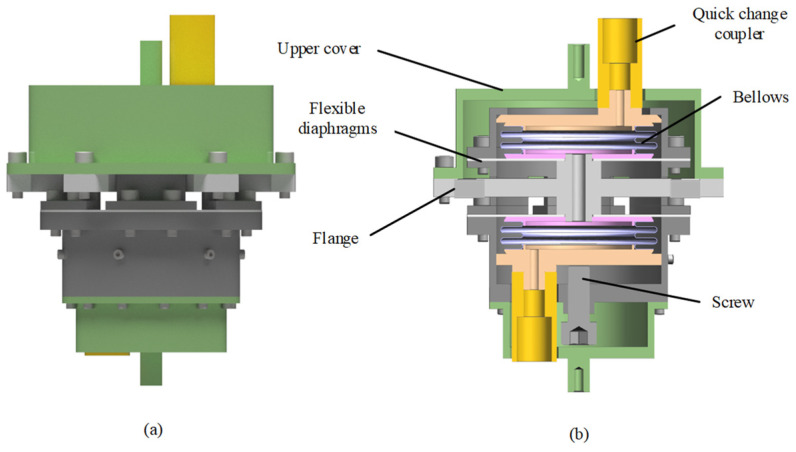
Schematic diagram of system structure: (**a**) overall; (**b**) sectional view.

**Figure 2 sensors-24-06265-f002:**
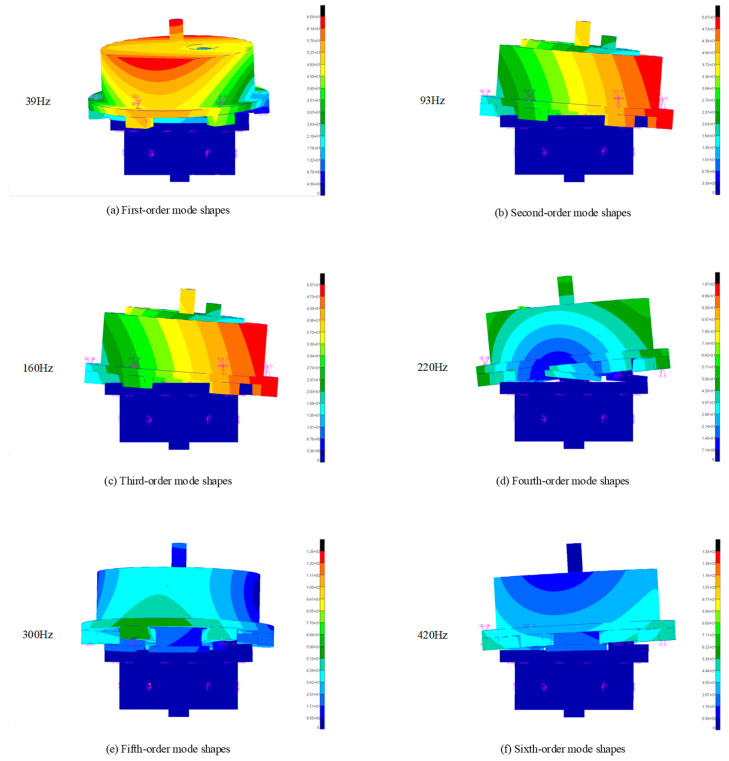
Sixth-order mode shapes in front of the damper.

**Figure 3 sensors-24-06265-f003:**
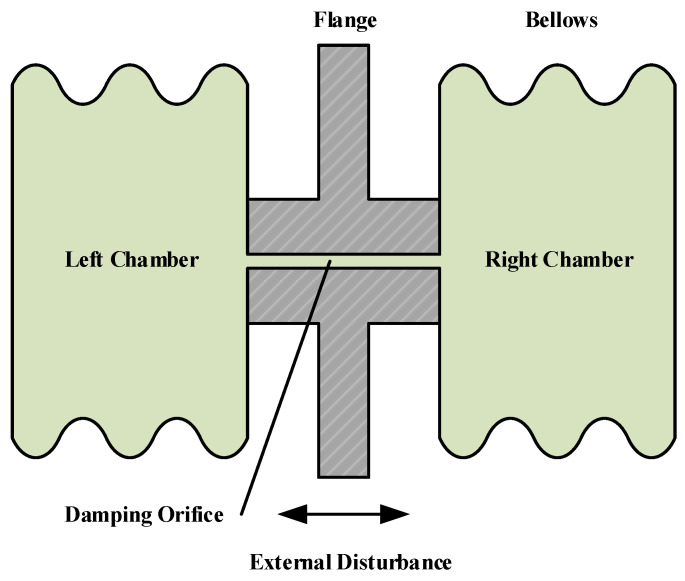
Schematic of a bellows-type FVD.

**Figure 4 sensors-24-06265-f004:**
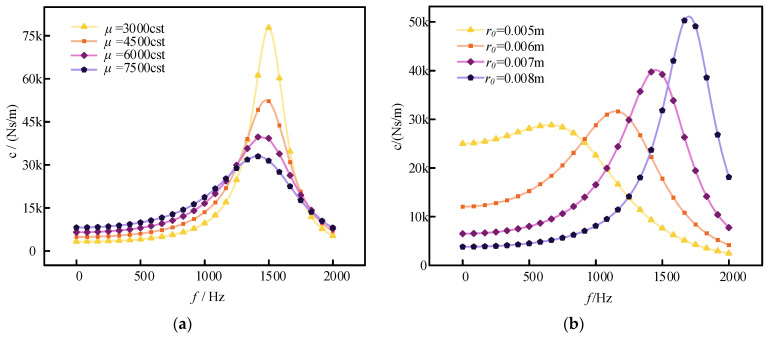
Impact of parameters: (**a**) damping coefficient at different viscosities; (**b**) damping coefficients at different apertures.

**Figure 5 sensors-24-06265-f005:**
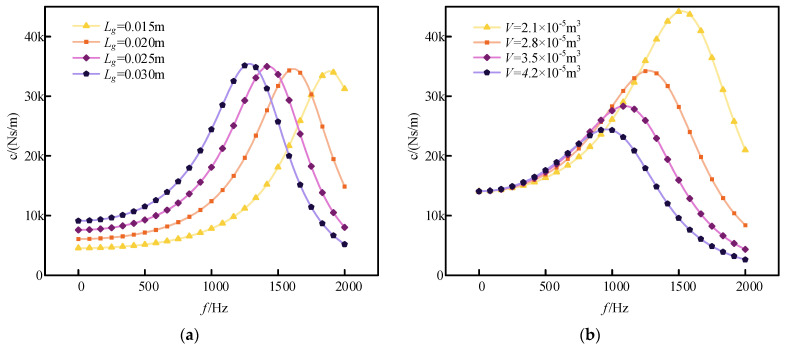
Impact of parameters: (**a**) damping coefficients for different hole lengths; (**b**) damping coefficients for different cavity volumes.

**Figure 6 sensors-24-06265-f006:**
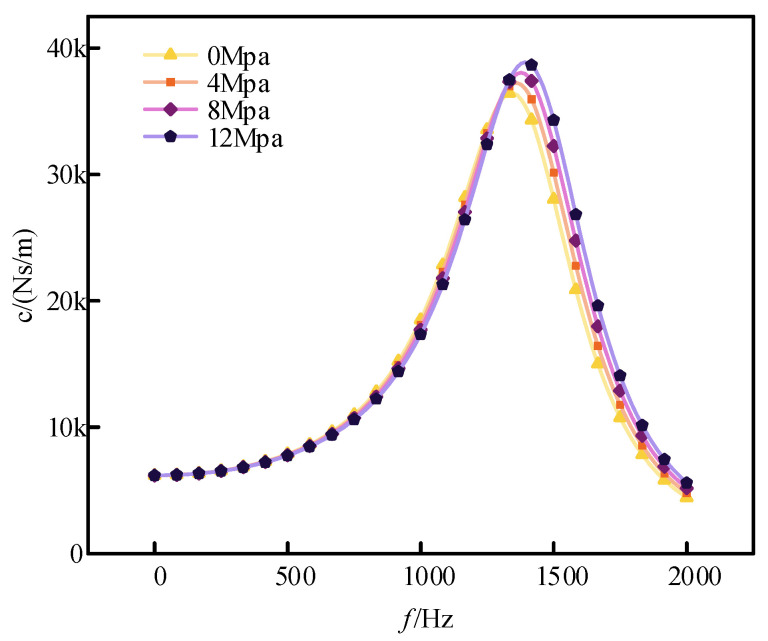
Damping coefficients at different cavity pressures.

**Figure 7 sensors-24-06265-f007:**
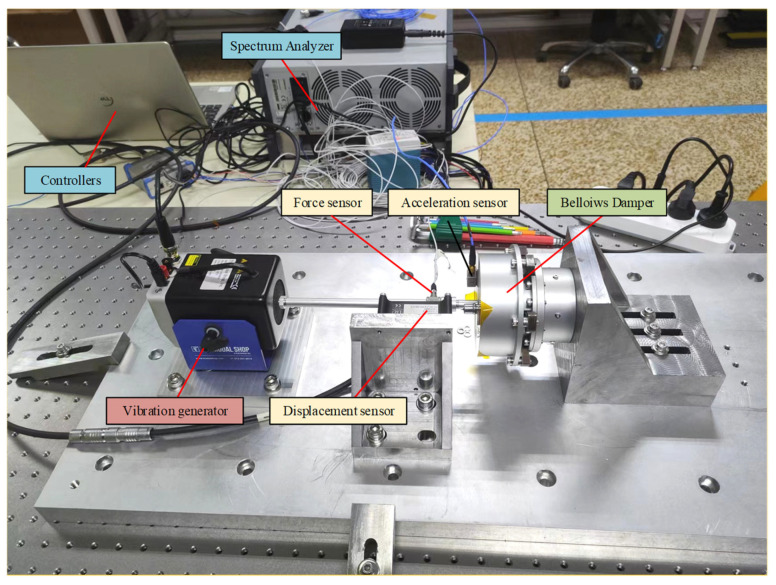
Experimental test platform.

**Figure 8 sensors-24-06265-f008:**
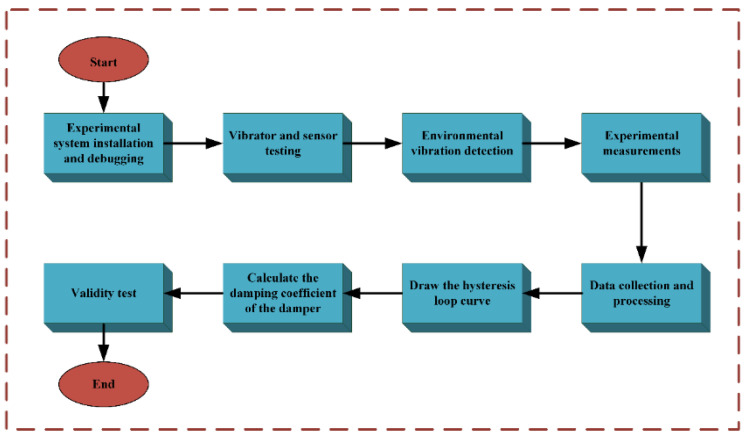
Experimental procedure.

**Figure 9 sensors-24-06265-f009:**
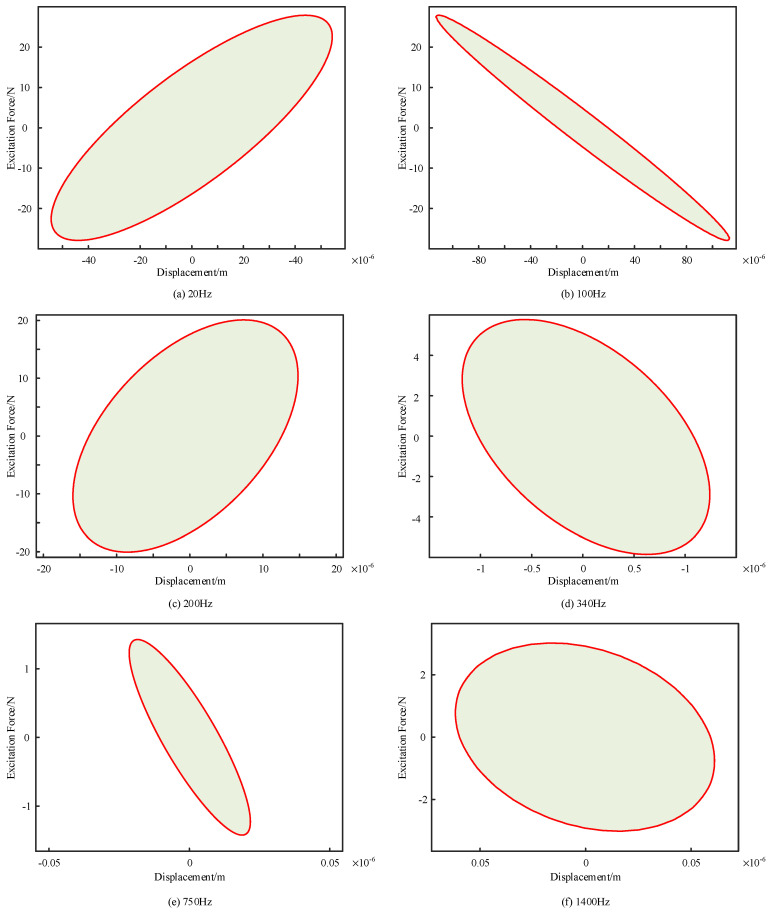
Hysteresis curves of dampers at different frequencies.

**Figure 10 sensors-24-06265-f010:**
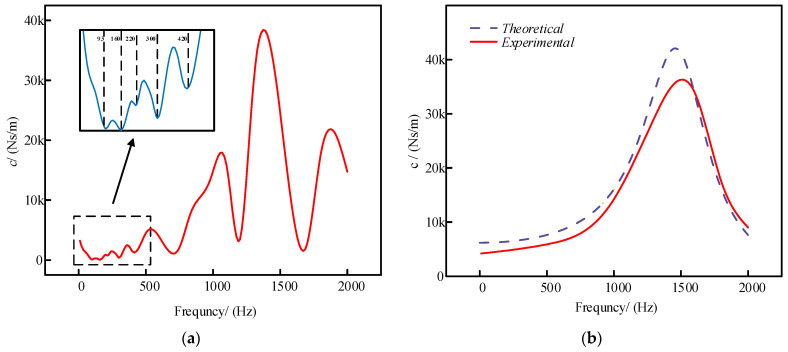
(**a**) Measured damping coefficient versus frequency curve; (**b**) data optimized damping curves and theoretical curves.

**Figure 11 sensors-24-06265-f011:**
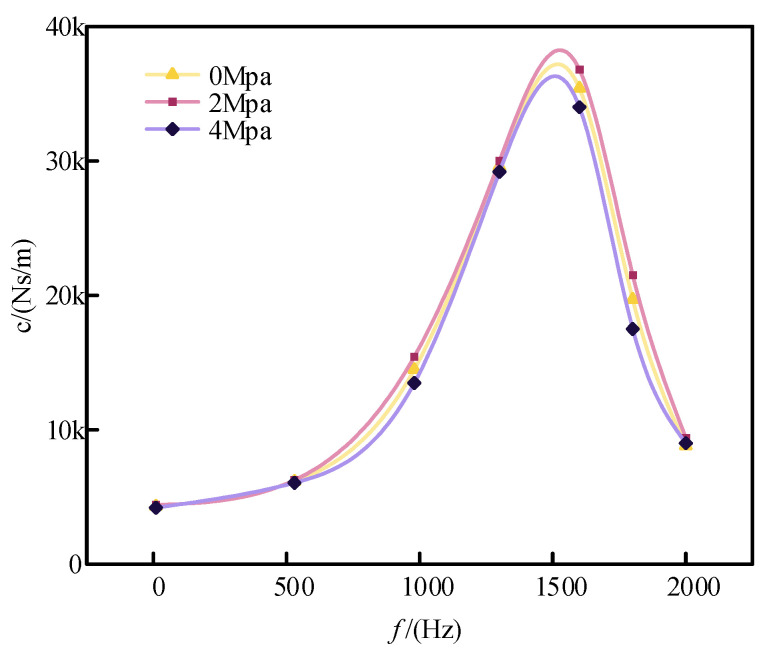
Damping versus frequency curves at different pressures.

**Table 1 sensors-24-06265-t001:** Theoretically calculated and practically tested damping peak magnitude at different pressures.

**Cavity Pressure** **Mpa**	**Experimentally Obtained Damping Peaks c** **(/Ns/m)**	**Theoretically Calculated Damping Peaks c** **(/Ns/m)**
0	35,048	36,411
2	36,475	36,734
4	37,846	36,989

## Data Availability

The data presented in this study are available on request from the corresponding authors. The data are not publicly available due to privacy restrictions.

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
