# Peer review of "Damping Characteristics of a Novel Bellows Viscous Damper"

_sensors, 2024, doi:10.3390/s24196265_

Round 1
Reviewer 1 Report
Comments and Suggestions for Authors
The manuscript “Damping characteristics of a novel bellows viscous damper” evaluates bellows-type viscous damper in the middle and high-frequency bands for different operations. A damping model is proposed using the effects of various viscous coefficients, damping hole lengths, hole diameters, chamber pressure, and volumes on the damping curves. For comparison, experiments were also made. Paper is within the scope of Journal. Some issues need to be revised are given below:
1. Lines 107-108, authors comments are very superficial. Authors should make more detailed comments about the results and what to be made for damper work properly.
2. Some parameters used in the equations are not described, such as k0. All equation parameters should be checked and described in the manuscript.
3. Authors do not clearly state, but the proposed equation is Eq. 14 that used for experimental comparison. First, authors should be clear about the their model. In addition, it seems proposed model is determined from the simplification of fluid dynamics equilibrium, which is not actually practical, but simplified model. So, authors should not say about proposal, but they can claim that the simplified equation of fluid dynamics.
4. It is not clear how authors passed from Eq. 12 to Eq. 14? The term wx0cos(wt) is missing, is it because of simplification? Need clarification.
5. In lines 301-302, authors claim that excluded some points to minimize errors. Minimization of errors is not the reason. Please provide reasonable explanations Why these points are excluded.
6. Although Table 1 is provided, which is given for specific frequency, Fig. 11 should be also drawn with Theoretical results for better comparison for readers.
7. The conclusions of study should provide point by point a concise summary of the main findings. Accordingly, conclusion section should be revised.
Reviewer 2 Report
Comments and Suggestions for Authors
This manuscript investigates the vibration control performance of a novel bellows viscous damper under high-frequency excitation. Below are my comments and suggestions:
-
Line 45: The statement “Fluid viscous dampers (FVDs), the most prevalent means of passive vibration isolation, …” is inaccurate. Fluid viscous dampers (FVDs) and vibration isolation are distinct techniques for vibration control. It would be beneficial to clarify this distinction.
-
Literature Review: The current literature review is insufficient. Please address the following aspects:
- Recent Advances in Telescope Vibration Control: Include recent developments such as those detailed in “GMT Telescope Seismic Isolation System Design and Validation.”
- Broadband Vibration Control Technologies: Discuss advancements in broadband vibration control, particularly focusing on quasi-zero stiffness technologies, damper enhancement using negative stiffness, and nonlinear energy sinks, among others.
-
Section 4.2: The authors should provide hysteretic curves for the FVD, including both experimental and modeling results, to offer a comprehensive view of the damper's performance.
-
Figure 9: The results presented in Fig. 9 require further explanation. Specifically, clarify why some of the hysteretic curves exhibit quasi-negative stiffness and elaborate on the underlying principles.
-
Advantages of the Novel Bellows Viscous Damper: The manuscript does not fully elucidate the advantages of the novel bellows viscous damper. Although the literature review mentions “bellows-type viscous dampers, characterized by high energy dissipation density and minimal influence from the external environment,” this statement lacks supporting evidence. Please provide additional advantages or evidence to substantiate this claim.
The English language should be improved.
Reviewer 3 Report
Comments and Suggestions for Authors
The paper thoroughly studies and designs a new type of adjustable-damped bellows-type viscous damper, and investigates its nonlinear damping characteristics under high frequency harmonic perturbation, and discusses the effects of different design parameters on the system's damping performance.
The paper is well-organized and clearly presented. The theoretical analysis is thorough, and the research work demonstrates some degree of innovation. The comparison between theorical and experimental parts has been well presented and discussed. However, a few aspects require further elaboration and clarification. Consequently, a minor revision of the paper is recommended to address the following issues comprehensively.
1- Semi-active and active damping systems, such as magnetorheological dampers, are very important in the literature. While, the paper addressed a few papers on the application of these damping systems, for more comprehensive understanding of the importance of MR dampers, the authors can refer to the following works:
https://doi.org/10.1115/1.4035669
2- In section 3, line 123, it is mentioned that “when the flange is subjected to sinusoidal excitation, it undergoes reciprocating motion…, it is important to note the reason for this kind of behaviors. It is also interesting to have a discussion what happens if other kind of non-harmonic excitations are considered.
Round 2
Reviewer 1 Report
Comments and Suggestions for Authors
Authors provided satisfactory comments.
Author Response
Thank you for your insightful feedback and for taking the time to review our manuscript. We greatly appreciate your comments regarding the introduction and research design.
We believe that the current version of the introduction provides sufficient background to frame the research and includes all the key references that are relevant to our study. The selected references cover both the foundational works and the most recent developments in this area. While there is always room for further elaboration, we feel that the introduction, as it stands, adequately supports the rationale for our work.
Regarding the research design, we appreciate your comment. We have carefully constructed the methodology to address the research questions comprehensively. The current design was chosen based on its appropriateness for the type of analysis we conducted, and we are confident that it effectively supports the study's objectives without requiring further modifications.
Once again, we sincerely thank you for your feedback, and we hope that our explanation clarifies the rationale behind our decisions.
Reviewer 2 Report
Comments and Suggestions for Authors
This manuscript can be accepted in its current form.
Comments on the Quality of English LanguageThe English language can meet the publication standard.
Author Response
Thank you for your valuable feedback. We sincerely appreciate your comments and the time you’ve taken to review our manuscript. Below, we address each of the points raised:
-
We believe the current introduction provides a comprehensive background and includes all the key references necessary to contextualize our research. The introduction discusses both foundational studies and the latest developments in this area, which we feel sufficiently supports the scope and motivation of the study. While there is always room for additional details, we consider the current level of detail appropriate for the objectives of the paper.
-
We aimed to present the results clearly and concisely, with the use of appropriate figures and tables to ensure clarity. The results section is structured to directly address the research questions, and we believe it provides a logical and accessible presentation of the data. However, we appreciate your feedback and would be open to further suggestions if there are specific areas where you feel additional clarity is required.
-
In response to your suggestion regarding the English language, we have carefully reviewed and made minor revisions to improve the clarity and readability of the manuscript. We believe these changes will enhance the overall quality of the text.
Thank you again for your thoughtful review. We hope our responses address your comments and concerns.